A comparison of least squares regression and geographically weighted regression modeling of West Nile virus risk based on environmental parameters

Kala Abhishek K. 1
Tiwari Chetan 2
Mikler Armin R. 3
Atkinson Samuel F. atkinson@unt.edu 1
1 Advanced Environmental Research Institute and Department of Biological Sciences, University of North Texas , Denton , TX , United States
2 Advanced Environmental Research Institute and Department of Geography and the Environment, University of North Texas , Denton , TX , United States
3 Advanced Environmental Research Institute and Department of Computer Science and Engineering, University of North Texas , Denton , TX , United States
Noymer Andrew
Electronic publication date: 2017 Mar 28
Publication date: 2017
Volume: 5
Electronic Location ID: e3070
Received 2016 Sep 8; Accepted 2017 Feb 7
Copyright: ©2017 Kala et al.
Copyright year: 2017
Copyright holder: Kala et al.
License: This is an open access article distributed under the terms of the Creative Commons Attribution License, which permits unrestricted use, distribution, reproduction and adaptation in any medium and for any purpose provided that it is properly attributed. For attribution, the original author(s), title, publication source (PeerJ) and either DOI or URL of the article must be cited.
License URL: https://creativecommons.org/licenses/by/4.0/

Keywords: Emerging infectious diseases, Avian impacts, West Nile virus, Spatial modeling, Geographic information systems (GIS), Model comparison

Funding: Advanced Environmental Research Institute University of North Texas Funding was provided by the Advanced Environmental Research Institute, University of North Texas. The funders had no role in study design, data collection and analysis, decision to publish, or preparation of the manuscript.

==============================
Background

The primary aim of the study reported here was to determine the effectiveness of utilizing local spatial variations in environmental data to uncover the statistical relationships between West Nile Virus (WNV) risk and environmental factors. Because least squares regression methods do not account for spatial autocorrelation and non-stationarity of the type of spatial data analyzed for studies that explore the relationship between WNV and environmental determinants, we hypothesized that a geographically weighted regression model would help us better understand how environmental factors are related to WNV risk patterns without the confounding effects of spatial non-stationarity.

Methods

We examined commonly mapped environmental factors using both ordinary least squares regression (LSR) and geographically weighted regression (GWR). Both types of models were applied to examine the relationship between WNV-infected dead bird counts and various environmental factors for those locations. The goal was to determine which approach yielded a better predictive model.

Results

LSR efforts lead to identifying three environmental variables that were statistically significantly related to WNV infected dead birds (adjusted R2 = 0.61): stream density, road density, and land surface temperature. GWR efforts increased the explanatory value of these three environmental variables with better spatial precision (adjusted R2 = 0.71).

Conclusions

The spatial granularity resulting from the geographically weighted approach provides a better understanding of how environmental spatial heterogeneity is related to WNV risk as implied by WNV infected dead birds, which should allow improved planning of public health management strategies.

Introduction

West Nile Virus (WNV) is a vector-borne disease that was first detected in the United States in 1999 (Nash et al., 2001). Within a few years the virus had spread across the North American continent (Hayes et al., 2005). WNV has had important environmental and human impacts, including a decline in numerous bird species (CDC) and increased morbidity and mortality among humans. This has also resulted in increased economic burdens due to initial acute health care needs of infected individuals and subsequent long-term costs associates with infection, estimated at approximately $56 million per year between 1999 and 2012 (Barrett, 2014). Because that study indicated how difficult predicting and planning for WNV outbreaks was, we became interested in developing a spatially explicit model using environmental factors in an attempt to improve WNV risk predictions.

There are two important considerations that should typically be examined when developing spatially explicit environmental disease risk models (Miller, 2012). The first should be an examination of potential spatial autocorrelation (the degree to which a set of spatial features and their associated data values tend to be clustered together in space). This involves accounting for whether environmental factors and the corresponding disease rates in geographically proximate areas are more or less clustered together than they are in geographically distant areas. Second, data non-stationarity (changing means, variances and covariances in data across space) should be investigated and controlled when necessary (Fotheringham, 2009a; Miller, 2012). Geographically weighted regression (GWR) can be used for these two considerations and can often produce improved models that enable better spatial inference and prediction. Recent studies have applied GWR modeling to drug-resistant tuberculosis versus risk factors (Liu et al., 2011); environmental factors versus typhoid fever (Dewan et al., 2013); local climate and population distribution versus hand, foot, and mouth disease (Hu et al., 2012); and environmental factors and tick-borne disease (Atkinson et al., 2012; Atkinson et al., 2014; Wimberly, Baer & Yabsley, 2008; Wimberly et al., 2008), all showing that predictor variables varied spatially across large geographic regions, implying that the results for such studies may be improved using GWR.

The spatially explicit model that is discussed in this paper uses GWR to account for spatial heterogeneity for two reasons: (a) WNV disease risk observed across space may be related to similar environmental variables that increase vector habitat suitability and (b) environmental variables that influence WNV risk are not typically uniformly distributed across geographic space. Although many epidemiological models of WNV risk have been developed, it appears that there has been little research to explicitly examine techniques that account for spatial heterogeneity. Most models assume that the impact of various environmental factors are constant across the study region, which is unrealistic as larger areas display substantial variations in distribution of environmental, socio-economic, and demographic conditions (Goovaerts, 2008).

Due to the unavailability of reliable and complete data, developing models of WNV risk pose additional challenges. Human case data is lacking due to issues of under-reporting and limited surveillance. Our alternative strategy was to assess WNV infected dead bird counts as a surrogate measure of human risk because “infection rates” in dead birds can be more precise because of the genetic markers tested in dead birds may be more reliable than case data and/or surveillance data. Additionally, others have also used mosquito habitat suitability as a surrogate for estimating WNV risk for human infection (Cooke, Grala & Wallis, 2006). For our study, we followed a similar approach and used a model of mosquito habitat suitability condition as a predictor of the spatial distributions of infected birds, which in turn can be used to estimate WNV disease risk among human populations. Further, because the environmental variables considered in this study are known to vary across space, we account for spatial autocorrelation and non-stationarity using GWR following the approach of (DeGroote et al., 2008) in order to improve the predictability of a model.

WNV transmission and risk factors

The WNV transmission cycle was an important component of our modeling efforts. The first step in the WNV transmission cycle primarily occurs when a competent female mosquito vector bites an infected bird reservoir host, which in turn results in the virus being transmitted to the mosquito (Blair, 2009). This occurs when the female mosquito is seeking a blood meal to obtain nutrients necessary for egg development. After taking an infectious blood meal, a mosquito may pick up a permanent infection. The infected mosquito now has the potential to transmit the virus to another bird or animal when it feeds again. Once infected, birds may fly to other locations where the virus can be transmitted to susceptible mosquitoes. Subsequently, the disease may be transmitted by infected mosquitoes to humans or other mammals that act as incidental hosts. Dead birds found to be infected with WNV are often the primary indicators for presence of the disease in a geographic region and have proven to be useful for disease prediction modeling and identifying areas for human infection risk (Cooke, Grala & Wallis, 2006; Ruiz et al., 2004; Valiakos et al., 2014). This relationship allows an assumption of a positive correlation between infected dead birds and WNV risk. Since the New York outbreak in 1999, WNV has been recovered from 26 mosquito species in North America, including Culex. pipiens, Culex. salinarius, Culex. restuans, Ochlerotatus canadensis, Oc. japonicus, Aedes vexans, and Culiseta melanura (CDC, 2000; Control and Prevention, 2001). Results from a study (Goddard et al., 2002) assesses the vector competence of California mosquitoes. The results indicate that mosquitoes in the genus Culex (Cx.) are the principal hosts of WNV in California. The study also analyzed that on the basis of vector competence and host-feeding patterns, Cx. tarsalis may be the principal vector in rural agricultural ecosystems; and Cx. pipiens complex and Cx. stigmatosoma as important vectors in urban settings.

Vector and pathogen reservoirs overlap when certain environmental conditions are present (Rochlin et al., 2011). Table 1 provides an overview of the environmental conditions that are associated with WNV transmission, which were utilized for our research. These include characteristics of a place such as the mosquito species habitat: climatic conditions, topography and land use/land cover classes such as vegetation, water, and urbanized areas. Spectral indices acquired from satellite imagery provide information about environmental characteristics like temperature, vegetation cover, and moisture (Rodgers & Mather, 2006). Liu & Weng (2012) in a study on WNV risk in southern California found that one of the main factors contributing to the WNV propagation included land surface temperature. They related higher temperature to viral replication in mosquitoes for WNV to be disseminated throughout the year. The results also show that areas with lower elevations tended to be more susceptible to WNV invasion as mosquito population propagates in the plain habitats with warmer temperatures compared to areas with higher elevation that have lower temperatures.

Table 1 Environmental conditions related to WNV transmission risk.

Factors studied	Relation to WNV	References	
Streams	Sites for breeding and resting	Cooke, Grala & Wallis (2006), Curtis et al. (2014) and Schurich et al. (2014)	
Temperature	Increases growth rate of vector, decreases egg development cycle and shortens extrinsic incubation period of vector	DeGroote, Sugumaran & Ecker (2014), Kuehn (2012), Srivastava et al. (2001) and Wimberly et al. (2008)	
Surface slope	Water stagnation creating mosquito breeding ground	Cooke, Grala & Wallis (2006), Ozdenerol, Bialkowska-Jelinska & Taff (2008), Schurich et al. (2014) and Srivastava et al. (2001)	
Cultivated land	Linkage between habitat used and human-commensal nature of WNV mosquito vectors	Kilpatrick (2011)	
Developed land	Linkage between habitat used and human-commensal nature of WNV mosquito vectors; warmer microclimates	Kilpatrick (2011)	
Roads	Sites for breeding and resting along roadsides	Cooke, Grala & Wallis (2006)	
Vegetation	Sites for breeding and resting.	Brownstein et al. (2002), Cooke, Grala & Wallis (2006), DeGroote et al. (2008), Ruiz et al. (2004), Schurich et al. (2014) and Srivastava et al. (2001)	
Evapotranspiration	Related to the amount of moisture that is related to mosquito abundance	Liu & Weng (2012) and Trawinski & Mackay (2008)	

Statistical Considerations

Miller (2012) suggests that a ‘global’ model is the one that assumes that the parameters (commonly mean and variance) of some process are constant across geographic space (commonly mean and variance), typically referred to as the spatial stationarity of a process. Miller suggests that in the case these parameters vary across geographic space (spatial heterogeneity), then such models may lead to inaccurate predictions and subsequent problems for decision-making. In an ecological context, spatial heterogeneity usually results from the interaction of various environmental processes that operate at different scales (Legendre, 1993). Fotheringham (2009b) used local statistics for linking the concepts of spatial autocorrelation and heterogeneity that are deemed important when developing spatial models. Local statistics disaggregate a global mean value into locally computed values for each spatial unit. It is based on a conceptualization of Tobler’s first law in Geography (Tobler, 1970) that specifies that “everything is related to everything else, but near things are more related than distant things.” Spatial autocorrelation is a commonly used measure of the degree of spatial heterogeneity.

GWR is a local regression method that can be used for diagnosing spatial heterogeneity between dependent and explanatory variables over space (Fotheringham, Brunsdon & Charlton, 2003). It is performed within local windows centered on the nodes of a regular grid. Each observation within the local window is weighted based on its proximity to the center of that window. This approach has several advantages: it avoids abrupt changes in the local statistics computed for adjacent windows, helps visualize spatial variability within the geographic entity, and allows analysis of regionally aggregated data (Goovaerts, 2008). A model’s predictive ability, particularly in ecological modeling, is influenced not only by the strength of relationships between the species and its environment, but whether the model recognizes if the relationships are operating at multiple spatial scales. GWR provides a framework for exploring scale-dependent effects. It tests the effect on a model’s predictive ability by systematically increasing the size of the local window (Miller, 2012).

GWR can be used for mapping the spatial distribution of a model’s coefficient values in order to identify potential missing variables or to suggest other underlying factors associated with the observed non-stationarity (Miller, 2012). GWR is also useful for exploratory data analysis and visualization; for example Kupfer & Farris (2007) used a ‘leave-one-out’ (jackknifing) methodology to compare residuals from GWR and ordinary least squares regression. They found that GWR often had more accurate predictions for sites that were difficult to predict (where both models had overall higher residuals), which is why we used a GWR framework for explicitly modeling the spatial relationships between WNV and its environmental risk factors.

Materials & Methods

Study area

The model was built for the state of California, which was the national epicenter of WNV activity in 2004 and 2005 (Jean et al., 2007). WNV was first detected there in July 2003 (Reisen et al., 2004). It is the third largest state by area in the United States and is made up of 58 counties. California has the largest population in the US, but it is unevenly distributed across the state. The state also has a variable landscape with a large valley in the middle, bounded by mountain ranges.

Environmental factors and data sources

Our model utilized various environmental factors (Table 1) that have been suggested as descriptive in local WNV risk distribution: surface slope, density of roads, density of streams, monthly mean temperature, monthly mean evapotranspiration, and land cover classes like vegetation, developed land, cultivated land, and open surface water. All environmental parameters except roads and streams (Table 2) were acquired in grid format and resampled to 120 m resolution as suggested by Cooke, Grala & Wallis (2006). Data resampling was done using the resampling tool available in ArcGIS software. The modeling method utilized in this study was based on analyzing data in raster format, and therefore road and stream vector data were converted to raster format using the ‘Kernel Density Estimation’ tool in ArcGIS to create road density and stream density grid files. The tool assumes a Gaussian distribution and thus assigns more importance towards the center of kernel in comparison to the features that are further apart.

Table 2 Data sources.

Data	Spatial resolution	Source	
Elevation	10 m	National Elevation Dataset (NED)	
LST	1 km	MODIS aboard the Terra and Aqua satellites	
NDVI	250 m	MODIS aboard the Terra and Aqua satellites	
Evapotranspiration (ET)	1 km	MODIS aboard the Terra and Aqua satellites	
Streams	Available in vector format	US bureau of reclamation	
Roads	Available in vector format	US Census bureau	
Cultivated land	30 m	National Land Cover Database	
Developed land	30 m	National Land Cover Database	
WNV infected dead birds count	County scale	USGS National wildlife health center	
WNV human incidence cases	County scale	USGS National wildlife health center	
Human population	County scale	US Census bureau	

Various dynamic environmental data including Normalized Difference Vegetation Index (NDVI), Land Surface Temperature (LST), and Evapotranspiration (ET) were downloaded from the Moderate Resolution Imaging Spectroradiometer (MODIS) toolbox incorporated in ArcGIS®. The Land Surface Temperature tool accesses MOD11-A1, the daily averaged LST product. The MOD11 product uses the algorithm which is optimally used to separate ranges of atmospheric column water vapor and lower boundary air surface temperatures into tractable sub-ranges. The NDVI is calculated according to the formula NDVI = (NIR − VIS)/(NIR + VIS) where NIR is the near-infrared radiance and VIS is observed radiation in the visible spectrum. NDVI data is available from either satellite with MODIS (Aqua or Terra) as a monthly average. The time lag between the hatching of a mosquito egg to an adult mosquito taking blood meals and becoming infected with WNV to the subsequent infection of a human and the appearance of WNV disease symptoms was taken into account and therefore environmental data used for this study was taken for the month of July, the month prior to peak WNV human incidence cases (Campbell et al., 2002).

Least squares regression (LSR) modeling

WNV disease annual incidence rate (cases per 100,000 populations) was used as the measure of disease severity in this study. Annual WNV-infected dead birds sentinel data, averaged for 2004–2010, was used as a surrogate of WNV risk and was the dependent variable for modeling purposes in this study because several other studies (Chaintoutis et al., 2014; Eidson et al., 2001a; Eidson et al., 2001b; Eidson et al., 2001c; Guptill et al., 2003; Johnson et al., 2006; Mostashari et al., 2003; Nielsen & Reisen, 2007; Patnaik, Juliusson & Vogt, 2007; Ruiz et al., 2004) have suggested links between infected dead birds and WNV human infection rates. Since wild birds are the primary reservoir hosts for WNV and indicator of human infection risk, we utilized this association to develop the disease prediction model. We determined the utility of this relationship by correlating the dead birds data with the human incidence rate (r2 = 0.409, determined by utilizing Ordinary Least Squares modeling within the Modeling Spatial Relationships tools within the Spatial Statistics Tools toolbox incorporated in ArcGIS® Arc Toolbox). While infected dead bird counts only explain about 40% of reported human cases in California, it is a highly significant predictor (p = 0.01). Hence we used a dead bird model, with infected dead birds as a dependent variable, to assess WNV risk among human population.

Interpretations of ordinary Least Squares Regression (LSR) model performance were based on assessing multi-collinearity, robust probability, adjusted R2 and Akaike’s information criterion (AIC) (Akaike, 1974). Multi-collinearity was assessed through the variance inflation factor (VIF) statistic, which measures redundancy among explanatory variables. Explanatory variables associated with VIF values larger than about 7.5 indicate that these variables are providing similar information, and they were removed one at a time from the model based on VIF value until the model became unbiased. Robust probability indicates the statistically significant variables that are important to the regression model. Examining VIF values and robust probability, we ran and re-ran LSR models until narrowing down to non-redundant and significant variables: land surface temperature; stream density, and; road density. Akaike’s information criterion (AIC) was then used to determine the best LSR model.

The next step was to explore GWR models that might better explain the variation in infected dead bird counts based upon environmental data. Spatial autocorrelation (Global Moran’s I) was utilized to assess whether the environmental factors exhibited a random spatial pattern (Goodchild, 1986), and where adequate models have a random distribution of the residuals (Mitchell, 2005).

Geographically weighted regression (GWR) modeling

Under conditions of non-stationarity in LSR modeling, geographically weighted regression (GWR) was explored to potentially improve modeling results. These results were determined by utilizing Geographically Weighted Regression modeling within the Modeling Spatial Relationships tools within the Spatial Statistics Tools toolbox incorporated in ArcGIS® Arc Toolbox. The same explanatory variables that were used in LSR modeling were used to run GWR rather than starting with the full global set of parameters so as to avoid introducing “improvement” that could not be attributed solely to which modeling approach was applied. In other words, if GWR modeling was not applied to the same variables as LSR modeling, but yielded better results, we would not know if the improvement was due to the modeling approach or the environmental data that was used to build each model.

Once key environmental factors were identified during LSR modeling, we proceeded to explore the spatial variability of local regression coefficients to determine whether the underlying process exhibited spatial heterogeneity (Fotheringham, Brunsdon & Charlton, 2003). A GWR local model was applied to analyze how the relationship between infected dead bird counts and environmental factors changed from one county to another. Unlike conventional LSR regression modeling, which produces a single regression equation to summarize global relationships among the independent and dependent variables, GWR detects spatial variation within relationships in a model and produces information useful for exploring and interpreting spatial non-stationarity (Fotheringham, Brunsdon & Charlton, 2003).

A spatial kernel was used to provide geographic weighting for the local window centered on the grid nodes used in our model. There are two possible categories of spatial kernels: fixed/adaptive and bandwidth, which is a key coefficient that controls the size of the kernel (Akaike, 1974). These kernels tend to be Gaussian or Gaussian-like which implies that distant samples are weighed lesser than the proximal ones. There are three potential bandwidth approaches: Akaike information criterion (AIC), cross validation (CV), and bandwidth parameter. For our GWR model, the AIC approach was chosen because the distribution of infected dead birds was not consistent in the study area. The following settings were used in ArcGIS GWR: bandwidth method = AIC and Kernel type = Adaptive.

Finally, we examined independency and normality of residuals, to evaluate the fit of the model. Local collinearity, the square root of the largest eigenvalue divided by the smallest eigenvalue, of our GWR model was also assessed but no data points were removed as they compromised model diagnostics. The adjusted coefficient of determination (Adjusted R2) was used for comparing LSR and GWR models to determine which approach would provide a better understanding of the relationship between environmental conditions and West Nile Virus risk (Fotheringham, Brunsdon & Charlton, 2003).

Results

LSR modeling identified land surface temperature (VIF = 1.046), stream density (VIF = 1.177), and road density (VIF = 1.143) as statistically significant (p < 0.05) variables related to WNV risk: (1) WNV  risk=−75.87+595.60RD+1.89LST−146.89SD

Where:

WNV risk = average infected dead bird count

RD = road density

LST = land surface temperature

SD = stream density.

The histogram of the LSR model’s residuals approximates that of a normal curve, with a non-significant (0.134, p > 0.05) Jarque–Bera statistic (Jarque & Bera, 1980), and the Moran’s I Index Z-score (1.23) all imply that the model is unbiased and significantly different than random.

However, the Koenker statistic (0.000007*, p < 0.05) confirmed non-stationarity in the LSR model indicating that there is not a consistent relationship between the explanatory variables and WNV risk across the study area. Further, the presence of mild heteroskedasticity was noted in the LSR model. We conclude that the LSR model is stable but non-stationary, suggesting that proceeding with GWR model was warranted.

The GWR model in this study was implemented using the following algorithm: (2) WNVrisk(i)=βi0+β(i1)RD(i)+β(i2)LST(i)−β(i3)SD(i)+ε(i)

where β coefficients are county (i) specific, and RD is road density, LST is land surface temperature, and SD is stream density.

Comparing the fit of the global LSR model (assumes homogeneity of variables across space) and local GWR model (makes no assumption of homogeneity), we found that the global LSR adjusted R2 was 0.61 (R2 was 0.66, P < 0.05, Fig. 1) with analysis run on all 58 counties. The local GWR adjusted R2 was 0.71 (R2 is 0.75, p < 0.05, Fig. 2) with a bandwidth of 54, which suggests that there has been some improvement by using a local modeling approach. Our preferred measure of model fit, AIC, gave a value of 567.7 for the global model and 551.4 for the local model. The difference of 16.3 is relatively strong evidence of an improvement in the model fit to the data. Further, the problem of hetroskedasticity that was noted in the OLS model was not observed in the GWR model.

Figure 1 Trendline plot for global LSR model (model: y = 0.6591x + 10.563; r2 = 0.66), dashed line ideal 1:1 relationship.

Figure 2 Trendline plot for local GWR model (model: y = 0.6911x + 10.259; r2 = 0.75), dashed line ideal 1:1 relationship.

We also tested the results using different bandwidth parameter. Several iterations were run but it was observed that although a smaller band-width criterion gave an improved combination of AICc and adjusted R2 values, it also compromised the model diagnostics by introducing local collinearity and thus instability in the model. Addressing local collinearity by removing the Counties having condition number greater than 30 affected the model’s overall results. Thus, it is better to have a larger band-width rather than violating model assumptions and to avoid the unstable prediction (Charlton & Fotheringham, no date; Nakaya, 2014).

Mapping the values of the standardized residual across California (Fig. 3A) provides a representation of: (a) areas with unusually high or low residuals and (b) whether the residuals were spatially autocorrelated. Counties with excessively large positive residuals would under-predict WNV risk, and counties with excessively large negative residuals would over-predict WNV risk. The spatial autocorrelation of GWR residuals for our model resulted in a Moran’s I value of −0.11 (p = 0.18), implying little evidence of any autocorrelation in them.

Figure 3 Spatial distribution of (A) standardized residuals; (B) land surface temperature coefficients; (C) road density coefficients; and (D) stream coefficients.

Local coefficient estimates for significant factors were mapped using quantile classification method. Figure 3B shows the variation in the model’s coefficient estimates for the land surface temperature (LST) variable. The map for the local coefficients reveals that the influence of this variable in the model varies considerably over California, with a strong north-south direction. The range of the local coefficient is from 1.26 for the northernmost counties to 3.06 for the southernmost counties—evidence that points to heterogeneity in the model structure within California. The global coefficient and all the local coefficients for this variable are positive—there is agreement between the two models on the direction of the influence of this variable. Figure 3C shows a similar distribution in north-south direction of positive road density coefficient. Figure 3D reveals the opposite for stream density coefficients, with larger values in the north and smaller values in south. Contrary to our initial thoughts, stream density demonstrated a negative relation to disease risk. This may reflect that flowing water is normally not suitable for larval development of the various species of mosquitos that commonly transmit WNV or that rasterizing the stream database into stream density introduces a component that is not yet fully understood.

Our best ordinary least squares model, the global LSR model (Eq. 1) produced an adjusted R2 of 0.61 (p < 0.05) with a corresponding corrected AIC of 567.70. Utilizing the same environmental variables, our best local GWR model (Eq. 2) produced an adjusted R2 of 0.71 (p < 0.05) with a corresponding corrected AIC of 551.4. A 16 point decrease in the AIC and approximately 16% improvement in the model performance suggest that incorporating spatial data improves the predictive ability of WNV risk.

Conclusions

One of the frequent technical issues in modeling disease risk is to incorporate local rather than global associations in these models (Foley, Charlton & Fotheringham, 2009). In spatial regression models, a global model can be used to examine the relationship between disease risk and potential explanatory factors which are based on the assumption that the relationship is a stationary spatial process (Miller, 2012). For a small and homogenous region of interest, it is reasonable to assume that the explanatory factors would not change significantly across the region, and the relationship between WNV risk and the potential factors would also be unchanged. However, important variables such as topography, climate, and population distribution change greatly when it comes to a large region like California with an area of over 163,000 square miles. California is geographically diverse and is equally varied in its range of climates with several climatic sub-regions recognized. It would be unexpected to find that the spatial stationarity assumption holds in such a large area.

The distinct north–south pattern revealed in our study could be attributable to typical latitudinal expressions of temperature and precipitation, especially since California has a north-south length of 1,350 km. This environmental pattern is also a likely contributor to the distribution of different mosquito species in the United Stated, especially notable in its manifestation in California. A recent report (CDC, 2013) shows that while Cx. Tarsalis is distributed throughout California, Cx. pipiens is a more important mosquito vector in northern California, while Cx. quinquefasciatus is more important in southern California. While WNV can be found in a wide variety of ecosystems, the north–south pattern of infected birds detected in this study may be expressed more noticeably in California due to the north–south differences in mosquito species distributions as observed in the Centers for Disease Control report.

Our results concur that understanding WNV risks is improved when considering spatial heterogeneity of the variables that affect the risk (Beck et al., 1994). Besides improving prediction accuracy, spatial heterogeneity can also provide insights into the underlying ecological processes controlling the distributions of vector populations and zoonotic pathogens (Wimberly, Baer & Yabsley, 2008) because GWR models consider spatial heterogeneity by separating the large heterogeneous region into smaller, more homogeneous local regions. Fotheringham (Fotheringham, 2009a), stated that an advantage of using GWR is that it accounts for much of the spatial autocorrelation in the residuals that is usually found in global modeling. Further, it is possible that a variable that is insignificant at the global level might be important locally.

There are several limitations of this study. First, it is assumed that factors suitable for mosquito habitat increase the likelihood of WNV spread in human populations. On the surface this seems to be reasonably apparent; however, we do not have specific evidence that this is true. Second, it is also assumed that the probability of human infection is higher in counties with multiple confirmed WNV bird cases, another reasonable conjecture with several references in the literature, but without direct confirmation. Potential problems with this assumption include varying human population density (e.g., two areas with the same number of infected dead birds reported but one area’s human population density is substantially different than the other), variations in level of public concern (as reports of infected dead birds increase, more people begin looking for dead birds), and resource availability might bias the reporting of dead birds (wealthy areas devote disproportionate resources to the issue). Thus, proper surveillance methods that take into consideration these limitations while collecting infected dead bird data will contribute to more meaningful results. Third, our approach assumes that people are infected within the county of their residence, ignoring the possibility of contracting an infection while traveling outside the county limits. Lastly, road density could also be correlated with dead bird surveillance effort and might be a potential bias for reporting dead birds. We recognize that if these assumptions do not hold, modeling WNV risks based on infected dead birds may yield biased results. However, if the assumptions do hold, the local modeling approaches should improve predictions of WNV risks.

The research described in this paper suggests that a spatially explicit local model using GWR approaches to adjust for spatial autocorrelation and non-stationarity can yield improved predictions compared to ordinary LSR modeling of WNV risk. A spatially explicit modeling technique may be useful in policy-making and decision-making depending on the granularity and resolution of available data. Identifying the spatial variations in relationships by estimating local regression parameters allows the spatial distribution and interaction of predictor variables to be explored. Analyzing local variations in relationships provides those concerned with public health policy the ability to target resources and to achieve improved outcomes through location-specific activities (Comber, Brunsdon & Radburn, 2011) because spatial heterogeneity can improve predictions by capturing geographic shifts in the ecological drivers (Wimberly, Baer & Yabsley, 2008). While environmental data used in this research were of fine resolution, WNV disease human incidence data and infected dead bird data that is used is available only at a coarser county scale. We had to assume that aggregating the environmental data up to the county adequately represented the environmental conditions presented in the county, but we knew that data aggregation was likely to introduce some uncertainty into the model. The dead bird model applied in this study can be used for better understanding of WNV risk and the techniques used could be replicated at finer spatial scales thus leading to better intervention efforts.

In summary, WNV, a globally emerging infectious disease, was found to be heterogeneously related to environmental factors at the county level throughout California during the time that our data were collected. Our findings may assist those conducting risk assessments for WNV transmission in local areas by helping local public health entities allocate resources and improve preparedness for an outbreak according to region-specific conditions.

The authors wish to thank the Center for Computational Epidemiology and Response Analysis, the Center for Remote Sensing, and the Advanced Environmental Research Institute at the University of North Texas for technical advice and access to software and hardware to undertake this project.

Additional Information and Declarations

Competing Interests

Author Contributions

Data Availability

The authors declare there are no competing interests.

Abhishek K. Kala conceived and designed the experiments, performed the experiments, analyzed the data, contributed reagents/materials/analysis tools, wrote the paper, prepared figures and/or tables, reviewed drafts of the paper.

Chetan Tiwari, Armin R. Mikler and Samuel F. Atkinson conceived and designed the experiments, performed the experiments, analyzed the data, contributed reagents/materials/analysis tools, wrote the paper, reviewed drafts of the paper.

The following information was supplied regarding data availability:

Kala, Abhishek; Atkinson, Sam; Tiwari, Chetan; Mikler, Armin (2017): Data and data dictionary—WNV Modeling. figshare. https://doi.org/10.6084/m9.figshare.4596859.v1.

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
