# Peer review of "A comparison of least squares regression and geographically weighted regression modeling of West Nile virus risk based on environmental parameters"

_PeerJ, doi:10.7717/peerj.3070_

## Round 0.1 · original submission · Major Revisions

A few comments:

- I expect a response document that addresses each and every point raised by the referees. However, my standard for "addressing" does not mean that you must slavishly adhere to every point.

In some cases, addressing a point may mean explaining why you chose not to implement a particular suggestion. Needless to add, I don't recommend this for all the suggestions.

- In order to gain acceptance, the paper must adhere to the journal's OpenData policy. This will include provision of a dataset in readable (e.g., comma-, whitespace- or tab-delimited ASCII) format, accompanied by at least a minimal codebook. As well as code you used to produce all the results.

·

Basic reporting

a. L88-102: Missing background on major mosquito vectors in study area

b. L115: Replace “varying” with “vary”

c. L146: Study area section would be more appropriate in the M&M

d. Figures 2&3: Trend line coefficients and sample sizes are not provided

e. L269-295: Much of the discussion section would be more appropriate in the results

Experimental design

a. L173: How was WNV disease annual incidence rate incorporated into the model? This measurement was mentioned, but not made clear with regards to how it contributed or pertained to the model.

b. L224-226: Counties were removed based on collinearity, but how many and which counties were removed?

Validity of the findings

a. L347-349: The findings of this paper seem to provide further evidence supporting the utility of GWR methods, but the stated contribution of the findings to local WNV understanding and public health management is not clear or well explained. If California counties already have dead bird data as a proxy for WNV risk, then how do the GWR findings presented contribute to local WNV risk assessment? This connection needs to be made clearer.

Additional comments

Overall, paper needs structural reorganizing.

Reviewer 2 ·

Basic reporting

The rationale for the project is clearly described. I would request that the authors give it another read through for grammar because I noticed a few typos and phrasing issues (see lines 66, 72, 77, etc.).

The paper is currently framed as a comparison of two model fitting methods – LSR and GWR. The paper finds that GWR is an improvement on LSR, which perhaps is not a very surprising conclusion. I would prefer to see more discussion of why these particular factors are important for WNV, and a short review of other studies that have looked at the environmental factors that drive variability in WNV in California and how their study’s results compare. A paragraph in the introduction and discussion on this topic would greatly enhance the general interest of this paper. For instance, why does stream density have a negative relationship with WNV-infected dead birds? I would have expected the opposite.

Experimental design

It seems the authors modeled only infected birds, and not West Nile virus risk for humans. This is not a problem, per se, but needs to be clearer in the title and throughout the paper. I was confused as I read, because WNV risk in humans is mentioned in the text several times, but I couldn’t find any evidence that it showed up as a dependent variable in any of the models. For example, e.g., Table 2 and lines 173-175, and 181-182 (“we … examine the extent to which infected dead birds are associated with WNV risk”), make it seem like there should be a model that includes human WNV incidence data, but I didn’t see one.

I disagree with the argument that infected bird data ‘fix’ the problems associated with human WNV case data (lines 76-80), because dead bird data have the same issues as the human incidence data: namely differences in surveillance effort. Additionally, the factors that influence infection of birds with WNV are fundamentally different from those that influence human infections (e.g., there could be a major hotspot of WNV infection in birds where no humans ever travel, and so its relevance for humans is minor). The most direct way to measure human WNV risk is human WNV infection data, and since that was not used, the authors should simply describe more clearly what was used. The paper is still useful as a paper about the risk factors of WNV in birds.

The authors need to provide more information in the methods about how the categorical factors in the original model were measured and defined. For instance, what does vegetation refer to in Table 1? How was it measured/defined? Same question for cultivated land. An extra column in table 1 with this information would be very helpful.

The methods also state (line 160) that factors were resampled to 120 m resolution. Please provide more specifics about how this was done, as this it not clear.

Why do the authors use Land Surface Temperature measurements instead of air temperature? It seems that air temperature measurements, such as those provided by Prism, would be more relevant.

Validity of the findings

Lines 204-207 Why do the authors start the GWR model from the results of the LSR model rather than beginning with the full global set of parameters? What if different predictors would be important due to spatial autocorrelation?

Figures 2-3 suggest there may be some heteroskedasticity in the residuals (less variance at higher values of dead birds). Also, wouldn’t this regression line be expected to fall on a one-to-one line? Please include a dashed one-to-one line for comparison.

Additional comments

Please comment in the discussion on the chance that road density could also be correlated with dead bird surveillance effort (e.g., more roads likely means more people, which may mean more people noticing dead birds).

I would be interested to hear the authors’ speculations about why the coefficient values change the way they do. They seem to show a clear North-South pattern for most of the factors, and I’d be curious to hear their take on why that could be. For instance, do they follow along distributional patterns of the different mosquito vector or bird host species?

Also, for figures 4-7, how were the cut-offs for the various color groupings decided? They are very different in range. For example for road density, the blue range is ~33, while the red range is 233.

Lines 173-175: what was WNV incidence used for if not as the dependent variable?

Figure 1 seems unnecessary, a simple statement about the normality of residuals would be sufficient

Figure 7 the coefficient range for stream density switches from negative to positive in the northern region. Any thoughts on why this would be? It seems a little surprising and potentially problematic. Why would stream density have a negative effect on WNV-infected dead birds? I would have thought more streams would improve mosquito habitat and increase WNV risk.

The authors might also consider grouping figures 4-7 into a single figure with a single legend and panels a-d, since they are so similar in style.

---

## Round 0.2 · Major Revisions

Where is the source code I asked for? I'm not seeing it. Forgive me if it's there but I missed it (if this is the case, please let me know exactly where to find it).

---

## Round 0.3 · Minor Revisions

The referees each have a small suggestion. One regarding what is probably a typo on line 287 and the other regarding a minor data labelling issue. Both should be easy (borderline trivial) to address.

·

Basic reporting

L 287: It seems that the inequality sign is reversed in that "...with a non-significant (0.134, p<0.05)..." should instead read "... with a non-significant (0.134), p>0.05)..."

Experimental design

No comment

Validity of the findings

No comment

Reviewer 2 ·

Basic reporting

N/A

Experimental design

N/A

Validity of the findings

N/A

Additional comments

This round of revisions appears to only address the open access sharing of the data and source code. My only comment with respect to these issues is that it is usually nice to include a metadata file explaining the columns in the dataset. For instance, in the file with human WNV cases, are those neuroinvasive or total?

---

## Round 0.4 · Minor Revisions

I cannot find the data dictionary in your zip archive. I am poised to accept this manuscript but I want to make sure everything is in order on the open-data aspects.

---

## Round 0.5 · accepted · Accept

Thank you for clarifying the data dictionary, etc. I am pleased now to accept this paper.